# Novel Germline *PHD2* Variant in a Metastatic Pheochromocytoma and Chronic Myeloid Leukemia, but in the Absence of Polycythemia

**DOI:** 10.3390/medicina58081113

**Published:** 2022-08-17

**Authors:** Aldesia Provenzano, Massimiliano Chetta, Giuseppina De Filpo, Giulia Cantini, Andrea La Barbera, Gabriella Nesi, Raffaella Santi, Serena Martinelli, Elena Rapizzi, Michaela Luconi, Mario Maggi, Massimo Mannelli, Tonino Ercolino, Letizia Canu

**Affiliations:** 1Department of Experimental and Clinical Biomedical Sciences “Mario Serio”, University of Florence, 50139 Florence, Italy; 2Medical Genetics, Azienda Ospedaliera di Rilievo Nazionale (A.O.R.N.) Cardarelli, Padiglione, 80131 Naples, Italy; 3Centro di Ricerca e Innovazione sulle Patologie Surrenaliche, AOU Careggi, 50139 Florence, Italy; 4European Network for the Study of Adrenal Tumors (ENS@T) Center of Excellence, 50139 Florence, Italy; 5Department of Health Sciences, University of Florence, 50139 Florence, Italy; 6Department of Experimental and Clinical Medicine, University of Florence, 50139 Florence, Italy; 7Endocrinology Unit, Azienda Ospedaliera-Universitaria Careggi, 50139 Florence, Italy

**Keywords:** germline variants, *PHD2* gene, metastatic pheochromocytoma, radiometabolic therapy, PPRT, chronic myeloid leukemia

## Abstract

Background: Pheochromocytoma (Pheo) and paraganglioma (PGL) are rare tumors, mostly resulting from pathogenic variants of predisposing genes, with a genetic contribution that now stands at around 70%. Germline variants account for approximately 40%, while the remaining 30% is attributable to somatic variants. Objective: This study aimed to describe a new *PHD2* (*EGLN1*) variant in a patient affected by metastatic Pheo and chronic myeloid leukemia (CML) without polycythemia and to emphasize the need to adopt a comprehensive next-generation sequencing (NGS) panel. Methods: Genetic analysis was carried out by NGS. This analysis was initially performed using a panel of genes known for tumor predisposition (*EGLN1*, *EPAS1*, *FH*, *KIF1B*β, *MAX*, *NF1*, *RET*, *SDHA*, *SDHAF2*, *SDHB*, *SDHC*, *SDHD*, *TMEM127*, and *VHL*), followed initially by SNP-CGH array, to exclude the presence of the pathogenic Copy Number Variants (CNVs) and the loss of heterozygosity (LOH) and subsequently by whole exome sequencing (WES) comparative sequence analysis of the DNA extracted from tumor fragments and peripheral blood. Results: We found a novel germline *PHD2* (*EGLN1*) gene variant, c.153G>A, p.W51*, in a patient affected by metastatic Pheo and chronic myeloid leukemia (CML) in the absence of polycythemia. Conclusions: According to the latest guidelines, it is mandatory to perform genetic analysis in all Pheo/PGL cases regardless of phenotype. In patients with metastatic disease and no evidence of polycythemia, we propose testing for *PHD2* (*EGLN1*) gene variants. A possible correlation between *PHD2* (*EGLN1*) pathogenic variants and CML clinical course should be considered.

## 1. Introduction

Pheochromocytoma (Pheo) and paraganglioma (PGL) are rare tumors with an incidence of 2–5 patients per million per year and an uncertain malignant potential [1]. Pheos represent about 80–85% of cases and PGLs represent about 15–20% [2]. The recommended diagnostic workup for suspected Pheo/PGL (PPGL) includes plasma free metanephrines and fractionated urinary metanephrines. Subsequently, computed tomography (CT) scans or magnetic resonance imaging (MRI) are used to locate the lesion [3]. Nuclear medicine exams such as ^123^I-meta-iodobenzyl-guanidine (MIBG) scintigraphy, ^18^F-fluoro-L-dihydroxyphenylalanine (^18^F-DOPA), and ^68^Ga-DOTATATE PET are predominantly used in patients with metastatic disease [4,5].

Surgery is the curative option in non-metastatic and metastatic PPGL, and in patients with metastatic disease, tumor resection has been reported to increase overall survival [6].

Medical therapy with alpha-blockers is necessary before surgery [7]. In patients affected by metastatic disease, ‘wait and see’ is the first option given the generally slow evolution of the disease [8]. Radiometabolic therapies with ^131^I-MIBG or peptide receptor radionuclide therapy (PPRT) have proved effective [9,10], while cyclophosphamide vincristine dacarbazine (CVD) is the chemotherapy most commonly administered [11]. Local therapy such as external radiation therapy, radiosurgery, radiofrequency, cryoablation, and ethanol injection should be considered for unresectable lesions [12]. Tyrosine kinase inhibitors, e.g., Sunitinib [13], Temozolomide [14], as well as immunotherapy such as Pembrolizumab [15], are novel medical options for metastatic PPGLs.

Up to 70% of PPGLs are caused by germline or somatic pathogenic variants in one of the known susceptibility genes [16,17]. Based on their transcriptional profile, PPGLs are classified into three clusters. Cluster 1 includes PPGLs with variants in genes encoding the hypoxia-inducible factor (*HIF*) 2α, the Von Hippel–Lindau tumor suppressor (*VHL*), the prolyl hydroxylase domain (*PHD*), fumarate hydratase (*FH*), and succinate dehydrogenase subunits (*SDHx*) [18,19]. These tumors are characterized by the activation of pseudohypoxic pathways and by an immature catecholamine-secreting phenotype [20]. Cluster 2 comprises PPGLs with pathogenic variants in the REarranged during Transfection (*RET*) proto-oncogene, the Neurofibromatosis type 1 (*NF1*) tumor suppressor gene, the TransMEMbrane protein (*TMEM127*) gene, the Harvey rat sarcoma viral oncogene homolog (*HRAS*) and the MYC Associated Factor X (*MAX*) gene. Cluster 2 PPGLs show activated MAPK and mTOR signaling pathways and are mostly benign exhibiting a mature catecholamine phenotype with a strong expression of phenylethanolamine N-methyltransferase (PNMT) [21,22,23,24]. Cluster 3 is the Wnt signaling cluster. These tumors are due to somatic mutations of the *CSDE1* gene or somatic gene fusions of the *MAML3* gene [2], but also patients with sporadic forms fall into this cluster. Cluster 3 tumors have a more aggressive behavior [19].

Due to the large number of known susceptibility genes, next-generation sequencing (NGS) technology is ideally suited for carrying out PPGL genetic screening.

The genetic screening proposed by Toledo et al. [25] includes the *PHD2* (also called *EGLN1)* gene (Egl-9 Family Hypoxia Inducible Factor 1). The encoded protein catalyzes the post-translational formation of 4-hydroxyproline in hypoxia-inducible factor (HIF) alpha proteins. HIF is a transcriptional complex, playing a central role in mammalian oxygen homeostasis, controlling energy, iron metabolism, erythropoiesis, development under hypoxic or pseudohypoxic conditions and mediating adaptive cell responses to these states. Under normoxic conditions, HIF is controlled by several enzymatic reactions, including prolyl hydroxylation by PHDs, leading to proteasome degradation. However, pseudohypoxia conditions can lead to HIF stabilization and transactivation of target genes [26,27].

Dysregulation of HIF contributes to tumorigenesis and cancer progression [28,29,30,31]. HIF also has a crucial role in the pathogenesis of neuroendocrine tumors, especially PPGLs, regulating the cluster 1 pathway. Furthermore, *PHD2* heterozygous variants cause polycythemia, supporting the importance of PHD2 in the control of red cell mass in humans [32,33] that may be associated with PPGL [34,35].

Here, we report a novel germline *PHD2* variant in a patient affected by metastatic Pheo and chronic myeloid leukemia (CML) in the absence of polycythemia.

## 2. Materials and Methods

Informed consent was obtained from the patient and the manuscript was written following the CARE guidelines.

### 2.1. Case Presentation

The timing of the clinical presentation of the present case is shown in Figure 1A.

A 17-year-old female was incidentally diagnosed in 2012 with a 2.7 cm left adrenal mass suggestive of adenoma. The patient came to our attention in 2014. The lesion had increased to 3.6 cm in size. Urine tests showed high levels of urinary Metanephrine (MNu: 488 mcg/day, normal value (nv) < 320) and urinary Normetanephrine (NMNu: 65,125 mcg/day, nv < 390), indicating Pheo. No alteration in urinary Methoxytyramine levels (MTXu: nv < 460) and blood count were observed (white blood cells 6.28 × 10^3^/µL normal value (nv) 4.00–11.00, red blood cells 5.09 × 10^6^/µL normal value (nv) 3.8–5.00, hemoglobin 14.5 g/dL nv 12.0–16.0, hematocrit 43.4% nv 35.0–48.8, platelets 287 × 10^3^/µL normal value (nv) 150–450).

The patient was subjected to left laparoscopic adrenalectomy in June 2014 and the post-operative course was uneventful. Histologic examination revealed Pheo (Figure 2A). There was no evidence of capsular or vascular invasion and necrosis. The mitotic count was lower than 3 per 10 high-power fields (HPFs) and atypical figures were not seen. A *Pheochromocytoma of the Adrenal Gland Scaled Score* (PASS) of 3 was assigned and the Ki67 proliferation index was <1%. A para-aortic lymph node was positive for the presence of chromaffin tissue (Figure 2B) and metastatic Pheo was diagnosed.

Post-operative urinary metanephrine levels remained high (MNu 51 mcg/day, NMNu 882 mcg/day, and MTXu 116 mcg/day). Subsequent controls revealed an increase in NMNu up to 1346 mcg/day in October 2015 with a negative CT scan and MRI. The patient underwent ^18^F-fluoro-dihydroxy-pheylalanine (^18^F-DOPA) positron emission tomography (PET) and ^68^Ga-DOTATATE (Figure 3A) that revealed an increased uptake of para-aortic and retrocrural lymph nodes. The disease remained stable with a progressive increase in urinary NMN levels up to 2391 mcg/day. In June 2020, on account of the tumor burden and increasing levels of urinary NMN, peptide receptor radionuclide therapy (PRRT) with ^177^Lu-DOTATATE was initiated after egg preservation. PRRT was interrupted after four cycles due to a rapid increase in platelets (up to 2348 × 10^3^/µL, nv 150–450). In January 2021,^68^Ga-DOTATATE showed a reduced tracer uptake (Figure 3B). Further investigations, including bone marrow biopsy and genetic analysis, were carried out. A standard karyotype identified the presence of the Philadelphia chromosome (BCR/Abl), the hallmark of CML, and therapy with Imatinib was prescribed (200 mg bid). The erythropoietin level rose mildly (34.7 mlU/mL, the normal value being 4.3–29). During therapy, red blood cell count and hematocrit and hemoglobin levels remained stable, and the last control resulted 3.70 × 10^6^/μL, 37%, and 12.2 g/dL, respectively. Mean corpuscular volume (MCV), mean cell hemoglobin (MCH), and ferritin were normal (85.3 fl, 28.5 pg and 13 ng/mL, respectively).

### 2.2. Genetic Analysis

Genomic and tumor DNA of the patient were extracted using the QIAsymphony CDN kit (Qiagen, Hilden, Germany). DNA quality and quantity were measured by Qubit ds HS Assay on Qubit 2.0 Fluorimeter (Thermo Fisher Scientific, Waltham, MA, USA).

According to the guidelines for genetic screening of PPGL, our patient was proposed for NGS targeting. Using the online DesignStudio software (Illumina, San Diego, CA, USA), probes were designed to cover the following genes: *PHD2* (*EGLN1*), *EPAS1*, *FH*, *KIF1B*β, *MAX*, *NF1*, *RET*, *SDHA*, *SDHAF2*, *SDHB*, *SDHC*, *SDHD*, *TMEM127*, and *VHL*, including exon–intron boundaries.

### 2.3. SNP-CGH Array

To exclude the presence of pathogenic Copy Number Variants (CNVs) and loss of heterozygosity (LOH) in tumor samples, Illumina Infinium 850k Bead Chip CGH/SNP Microarray was performed according to the manufacturer’s instructions.

### 2.4. Whole Exome Sequencing and Bioinformatics Analysis

We decided to perform whole exome sequencing (WES) on blood and tumor DNA to sequence all coding genes in order to detect other variants in genes not included in our panel.

To construct DNA libraries, we used a strategy based on enzymatic fragmentation to produce dsDNA fragments followed by end repair, A-tailing, adapter ligation and library amplification (Kapa Biosystems, Wilmington, MA, USA). Libraries were hybridized with the protocol SeqCap EZ Exome v3 (Nimblegen, Roche, Basel, Switzerland) and sequenced by NextSeQ550 platform (Illumina Inc., San Diego, CA, USA).

### 2.5. 3 D Variant Prediction

To reveal the possible structural consequences of the identified variant, a 3D model of the truncated PHD2 protein was generated using the Phyre2 (Protein Homology Fold Recognition Engine) server created by the Structural Bioinformatics Group, Imperial College, London. Phyre2 uses the alignment of hidden Markov models via an HH search—an open-source software program for protein sequence searching—to significantly improve alignment accuracy (http://www.sbg.bio.ic.ac.uk/~phyre2/html/page.cgi?id=index, accessed on 13 June 2022) [36]. This was followed by I-TASSER (Iterative Threading ASSEmbly Refinement—https://zhanglab.ccmb.med.umich.edu/I-TASSER/, accessed on 13 August 2022) [37] to evaluate function the predictions and possible interactions of truncated PHD2 protein with other proteins and GRAMM v1.03, a program for protein docking to predict the structure of possible complexes (http://vakser.compbio.ku.edu/resources/gramm/grammx/, accessed on 13 August 2022) [38,39]. The generated *.pdb files were loaded and visualized with ChemDraw software to envisage a 3D structure (version 8; Cambridge Software; PerkinElmer, Inc., Waltham, MA, USA).

### 2.6. PHD2 Immunohistochemistry

PHD2 expression was assessed by probing formalin-fixed and paraffin-embedded tissue sections with mouse monoclonal antibody anti-PHD2 at 10 µg/mL (Abcam Cat#ab103432, RRID: AB_10710680). This antibody specifically recognizes a synthetic peptide corresponding to amino acids 1–24 of human PHD2 and is therefore capable of recognizing both the *wild-type* (*wt)* protein and the mutant/truncated form. Antigen retrieval was achieved using Epitope Retrieval Solution Citrate buffer (10 mM, pH 6; Dako, Glostrup, Denmark) in a thermostatic bath. Immunohistochemical analysis was performed using EnVision FLEX Systems and 3,3′-diaminobenzidine as the chromogen in a Dako Autostainer Link48 Instrument (Dako). Negative controls were incubated without the primary antibody. The sections were lightly counterstained with Mayer’s hematoxylin and mounted with Permount.

### 2.7. Western Blot Analysis

Dissected tumor tissues or human healthy adrenal samples (100 mg) were chopped in lysis buffer, incubated for 30 min on ice, and centrifuged at 10,000× *g* for 15 min at 4 °C. Proteins were quantified by Coomassie Blue-reagent (Bio-Rad, Hercules, CA, USA) [40] and 40 μg of proteins was separated by SDS/PAGE then transferred onto PVDF (Immobilon, Millipore, Burlington, MA, USA), as previously described [41]. Bound antibodies detected by ECL reagents (Immobilon, Millipore, Burlington, MA, USA) were analyzed with a Biorad ChemiDoc Imaging System (Bio-Rad, Quantity-One Software). HIF2α polyclonal antibody was supplied by Novus Biologicals (Bio-Techne, Minneapolis, MN, USA), while anti-GAPDH monoclonal antibody and anti-rabbit and anti-mouse secondary antibodies conjugated to horseradish peroxidase were from Santa Cruz Biotechnology (Santa Cruz, CA, USA).

### 2.8. RNA Isolation and Quantitative Real-Time PCR

Five different sample tissues obtained from pheochromocytoma (PHEO) and two from paraganglioma (PGL) were lysed for mRNA extraction. mRNA was isolated from frozen tissue using the RNeasy Mini Kit (Qiagen, Hilden, Germany), as previously described [42,43].

For each RNA sample, cDNA was obtained by reverse transcription PCR starting from 250 ng of RNA in 50 μL final volume reaction (Taqman RT-PCR kit; Applied Biosystems, Foster City, CA, USA) through the following cycling conditions: 10 min at 25 °C, 30 min at 48 °C, 3 min at 95 °C, and then held at 4 °C. Further quantitative real-time PCR (qRT-PCR) was carried out using primers and probes from Applied Biosystems for the gene transcripts human *PHD2* (Hs00254392_m1) and *GAPDH* (4352934). RT-PCR reactions were performed in triplicate for each gene on an ABI Prism 7900 Sequence Detector (Applied Biosystems). The number of target genes, normalized to the endogenous reference gene (human *GAPDH*) and relative to a calibrator (Stratagene, San Diego, CA, USA), was calculated by 2^−ΔΔCt^.

## 3. Results

The flow chart illustrating the filtering process and the variant selection used to identify the pathogenic variants is shown in Figure 1B.

### 3.1. Mutational Analysis

Analysis of SNP-CGH array reveals the absence of pathogenic CNVs and LOH in the entire genome. In particular, we focused on chromosome 1 to identify or exclude rearrangement or LOH near *EGLN1* that, in combination with the nonsense variant in *EGLN1*, could have some effect on our patient’s phenotype.

NGS identified a heterozygous c.153G>A (p.W51*) variant in exon 1 of the *PHD2* gene (NM_022051.3) which has been defined as pathogenic according to the American College of Medical Genetics (ACMG) guidelines. This variant is not reported in GnomAD, ExAC, or dbSNP NFE (European non-Finnish) databases. *PHD2* variant entails a G to A transition of the TGG coding triplet in a stop codon, which encodes for a truncated PHD2 protein (Figure 4).

Its pathogenicity is presumed because it leads to a premature stop codon and the bioinformatic tool (http://autopvs1.genetics.bgi.com/, accessed on 13 August 2022) predicted mRNA decay with a strong probability [44].

WES comparative sequence analysis of the tumor and blood DNA confirmed the variant identified by the panel genes, but no other significant variants were detected in the other coding regions.

Computational analysis highlights the possible involvement of a peptide in two different mechanisms: reduction in propensity of PHD2 to generate dimers and straight interaction with nuclear receptor corepressor 2 (NCoR2). Firstly, four possible amino acid regions have been detected that, interacting with truncated a peptide, could interfere with the formation of a disulfide bond (Figure 5A–D). These Aa regions are located among the DSBH domain (aa 299–390 light blue) which possesses the three cysteine residues (Cys302, Cys323, and Cys326 red) required for oxidative dimerization, through the formation of the disulfide bond and/or induction of a conformational change after oxidation.

Secondly, a potential interaction of a truncated peptide with a specific Aa sequence “TISNPPPLISSAK” of NCoR2 protein in positions 1101–1013 was identified. This NCoR2 sequence extends to the third region of the RII domain (residues 752–1016) and interacts with HDAC3, a Class I member of the histone deacetylase superfamily (Figure 5F).

Genetic analysis was extended to the patient’s parents with their consent. Genomic DNA was screened for the *PHD2* variant using PCR and Sanger sequencing. The patient had inherited the *PHD2* variant from her father.

### 3.2. PHD2 and HIF Expression in the Primary Tumor

Compared with non-PHD2-mutated Pheo and PGL, *PHD2*-mutated tissue showed a significant reduction in gene expression levels of *ENGL1* mRNA of 43% and 47%, respectively (Figure 5E).

Neoplastic cells were negative for PHD2 expression. Strong PHD2 staining of endothelial cells was evident in the lining of large blood vessels as well as in the capillaries. PHD2 expression was restrained to the cytoplasm (Figure 6). In comparison, the tumor expressed higher levels of HIF2α than a healthy adrenal, as shown by Western blot analysis (Figure 7).

## 4. Discussion

Here, we describe a novel nonsense *PHD2* germline variant in a patient affected by metastatic Pheo and CML with no evidence of polycythemia.

The first case of a *PHD2* missense variant observed in a 30-year-old patient with polycythemia and abdominal PGL was reported in 2008 [34]. In 2014, a novel *PHD2* germline missense variant was detected in a patient affected by Pheo [45]. More recently, a novel *PHD2* germline missense variant was found in a patient with PPGL and polycythemia [35]. None of these patients presented metastatic disease.

In contrast, nonsense variants in the *PHD2* gene were exclusively described in patients over 35 years of age suffering from polycythemia without chromaffin diseases [46,47]. We cannot exclude that in our case the absence of polycythemia could be explained by the younger age of the patient.

The *PHD2* variants so far described in association with polycythemia are all heterozygous [48], suggesting that a partial loss of PHD2 activity is sufficient to induce polycythemia. Although only a few of the *PHD2* variants have been reported, including those associated with tumors other than Pheo, a larger number of patients are needed to understand the exact function of heterozygous variants.

The *PHD2* gene encodes prolyl hydroxylase domain-containing protein-2 (PHD2), catalyzing the post-translational modification of hypoxia-inducible transcription factors that play an essential role in oxygen homeostasis. Prolyl hydroxylation is a basic regulatory event that targets HIF subunits for proteasomal demolition via the Von Hippel–Lindau ubiquitylation complex [47]. At the physiological oxygen level (normoxia), PHD hydroxylates proline residues on HIF-α subunits leading to their destabilization by promoting ubiquitination via the Von Hippel–Lindau (VHL) ubiquitin ligase and subsequent proteasomal degradation. In hypoxia, the O_2_-dependent hydroxylation of HIF-α subunits by PHD is reduced, resulting in HIF-α accumulation, dimerization with HIF-β, and migration into the nucleus to induce an adaptive transcriptional response. Variants in the *PHD2* gene determine a pseudohypoxia status. Indeed, PHD2 is unable to hydroxylate HIF-α, with the consequent stabilization of HIF-α subunits which are recognized as contributing to the pathogenesis of hereditary PPGLs [48].

As expected, our patient showed a lower expression of PHD2 and higher levels of HIF2α compared to the healthy adrenal tissues, confirming that PHD2 down-regulation results in HIF2α stabilization [49].

In our patient, we identified the new variant (c.153G>A) in the *PHD2* gene that introduces a stop codon, resulting in the production of a truncated protein which could explain the unexpected clinical status.

This variant is not reported in any SNV database. The gene constraint (gnomAD) indicates a strong probability of being LoF intolerant (oe-score: 0.06); therefore, it was considered pathogenic. In addition, we did not identify any exonic/splicing variants in the other coding regions.

We also evaluated CNVs/LOH through SNP-CGH arrays, which allowed for the exclusion of any possible genomic rearrangements and also reinforced data from the literature reporting that haploinsufficiency and partial deregulation of *PHD2* is sufficient to cause polycythemia [49].

Computational analysis indicated two possible effects of the truncated peptide. Firstly, an interaction between the truncated peptide and wild-type PHD2 protein was hypothesized. The truncated protein reduces the propensity to generate PHD2 dimerization-blocking establishment of the disulfide bond among DSBH domains and subsequently hindering HIF-1α activation by increasing glucose flux and lactate production (the Warburg effect) under oxidative stress. This interference could have a protective effect, decreasing tumor growth, as seen in our patient. Secondly, the interaction and inhibition of the transcriptional regulation of NCoR2 were considered, which would lead to the activation of HDAC3 and repression of nuclear receptors such as the thyroid hormone receptor and the retinoic acid receptor. HDAC3 is recruited by enhancers to modulate both the epigenome and nearby gene expression and is the only endogenous histone deacetylase that has a unique role in modulating the transcriptional activities of nuclear receptors.

Moreover, the heterozygous nonsense variant in the *PHD2* gene, as well as the residual mRNA produced by the wild-type allele, may act in a dominant-negative fashion to lower protein activity, possibly leading to a non-canonical-associated phenotype with metastatic Pheo, but with no clinical signs of polycythemia.

At genetic analysis, the patient’s father was positive for the known *PHD2* variant. CT scan and metanephrine assays proved negative. To date, considering the rarity of *PHD2* variants, the penetrance of this variant is unknown. An incomplete penetrance could be assumed, such as that of *SDHB* germline variants (9–75%) [50]. Indeed, in the previously reported cases, it was not possible to study the transmission of the disease [34,35,45].

On the other hand, autosomal dominant inheritance has been reported in patients affected by *PHD2* variants and polycythemia [51].

In summary, the role of *PHD2* heterozygous state as the driver gene associated with an unusual phenotype could depend on its relative abundance in a specific tissue, its interplaying among the three isoforms, as well as other genetic and epigenetic mechanisms.

We can neither exclude the presence of other variants in the non-coding or regulatory regions nor epigenetic alteration that, in association with the nonsense variant in the *PHD2* gene, may contribute to the development of the patient’s phenotype.

In agreement with the literature [52], we demonstrated that the *PHD2* variant led to HIF stabilization and its consequent activation through a significant increase in HIF2α expression in the mutated tumor. Focusing on HIF activation, we suggest a possible correlation between HIF upregulation and response to CML therapy. Considering the presence of the Philadelphia chromosome, we have not assumed a possible relationship between PPRT therapy and CML. In the literature, there are no data on this correlation.

In CML cell populations, HIF-responsive genes are upregulated by BCR/Abl [53]. Under low oxygen conditions, HIF supports the maintenance of stem cell potential, promoting the expansion of the mutated progenitor and an increased production of BCR/Abl protein [54]. This mechanism could be maintained in pseudohypoxia conditions. In CML murine models, HIF-1α genetic knockout prevents CML development by impairing cell cycle progression and inducing apoptosis in leukemia stem cells (LSCs) [55]. Therefore, HIF-1α is a critical factor in CML. LSCs, selected under low oxygen tension, are tyrosine kinase inhibitor (TKI)-insensitive [56] and HIF-1α-dependent signaling is relevant to LSC maintenance in CML [57]. This establishes HIF targeting as a possible strategy for CML treatment in patients who are insensitive to Imatinib and other TKI inhibitors [58]. Our patient was given Imatinib only recently. Future studies are necessary in order to verify the response to TKI inhibitors in *PHD2*-mutated patients. The description of an unusual clinical feature through NGS DNA analysis can significantly increase the diagnostic rate and improve patient management regardless of specific clinical signs.

## 5. Conclusions

We report a novel *PHD2* nonsense germline variant in a patient affected by metastatic Pheo and CML in the absence of polycythemia. Our findings confirm the need to screen patients affected by chromaffin disease using a comprehensive NGS panel, with no limitation regarding phenotype. In particular, we suggest extending *PHD2* gene analysis to younger non-polycythemic patients.

## Figures and Tables

**Figure 1 medicina-58-01113-f001:**
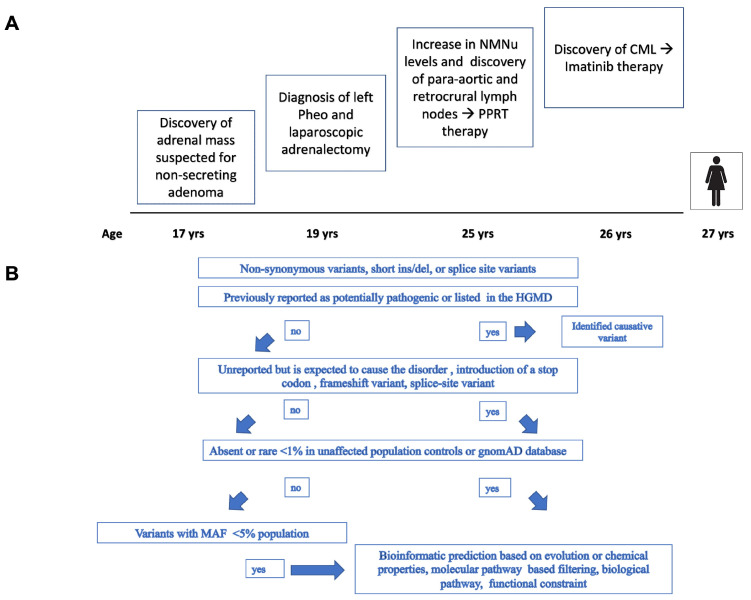
(**A**): Timeline of clinical presentation of the present case. (**B**): Flow chart illustrating the filtering process and variant selection used to identify pathogenic variants.

**Figure 2 medicina-58-01113-f002:**
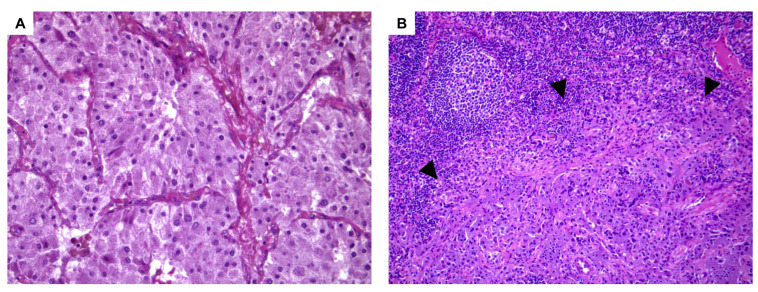
Histologic examination. (**A**): Pheochromocytoma showing nested architecture around a rich vascular network (H&E, original magnification ×20). (**B**): Nodal metastasis from pheochromocytoma is indicated by arrowheads (H&E, original magnification ×10).

**Figure 3 medicina-58-01113-f003:**
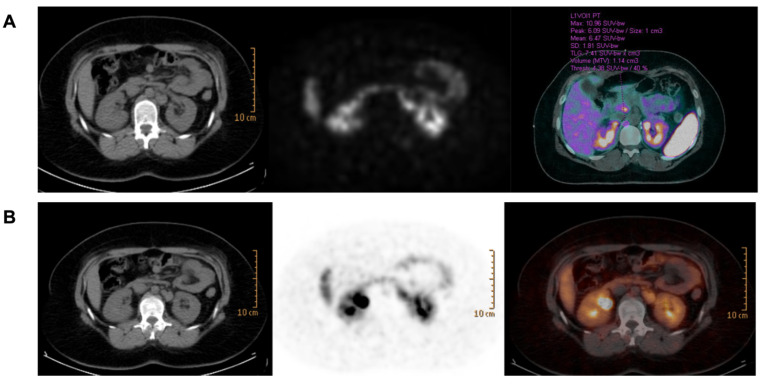
(**A**): 68Ga-DOTATATE pre therapy (21 February 2019): retrocrural right lymph node (SUVmax 11), posterior left renal artery lymph nodes (SUVmax 19.6), and retropancreatic lymph node (SUVmax 10.7). (**B**): 68Ga-DOTATATE pre therapy (28 June 2021): retrocrural right lymph node (SUVmax 6), posterior left renal artery lymph nodes (SUVmax 8.9–4.8), and retropancreatic lymph node (SUVmax 6.5).

**Figure 4 medicina-58-01113-f004:**
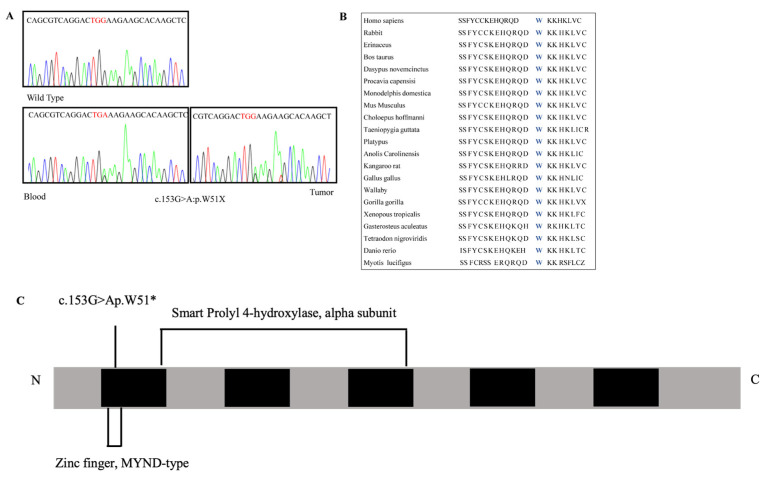
Genetic analysis. (**A**): Wild-type and mutated electropherogram showing the variant c.153G>A, p.W51* variant in exon 1 of the *EGLN1* gene in both blood and tumor samples. (**B**): Amino acid conservation among the species. (**C**): Scheme of *ENGL1* gene. Five boxes indicate the *ENGL1* exons (black).

**Figure 5 medicina-58-01113-f005:**
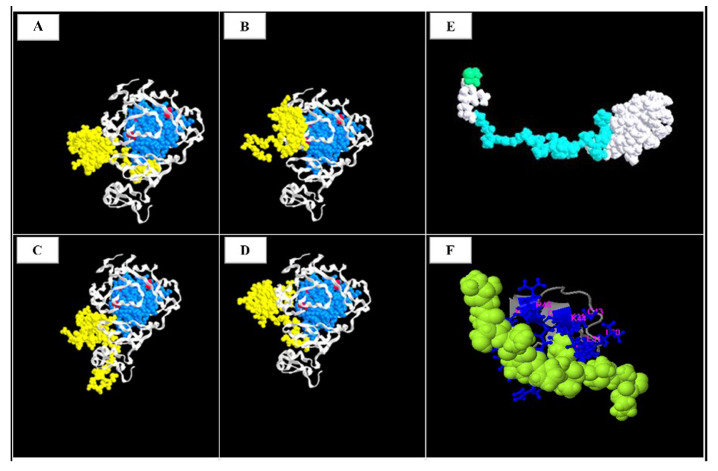
Computational analysis. (**A**–**D**): Possible interaction between peptide and wt PHD2; (**E**): Three-dimensional model of truncated PHD2 protein; (**F**): NCOR2-truncated PHD2 interaction.

**Figure 6 medicina-58-01113-f006:**
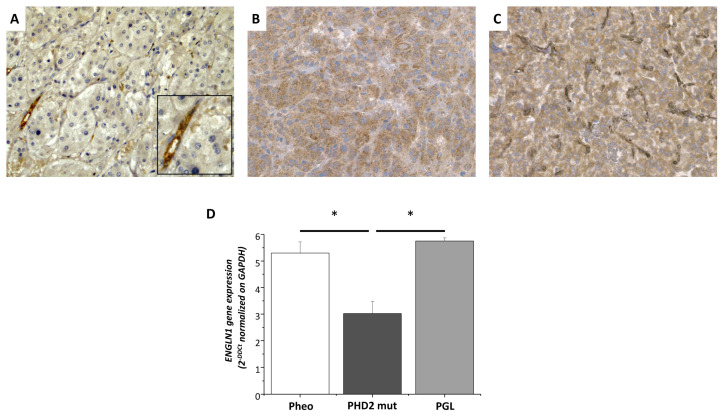
Immunohistochemical and QRT-PCR analyses for PHD2 expression. Patient neoplastic cells were not immunoreactive for anti-PHD2 antibody. Inset: strong PHD2 positivity was evident in the endothelial cells lining intra-tumoral capillaries ((**A**), original magnification ×20). Tumor cells showing granular, cytoplasmic staining for PDH2 in pheochromocytoma wild type for Krebs cycle genes ((**B**), original magnification ×20), and paraganglioma with *SDHD* gene pathogenic variant ((**C**), original magnification ×20). Five pheochromocytoma samples characterized by different mutational profiles (2 wt, 1 mutated *NF1*, 1 mutated *RET*, 1 mutated *PHD2*) and two PGL samples (1 wt and 1 *SDHB*-mutated) were subjected to quantitative RT-PCR. Data were expressed as mean ± SE of the *EGLN1* gene expression level normalized to the *GAPDH* expression in 3 independent experiments. * *p* < 0.01 vs. *PHD2*-mutated tissue (**D**).

**Figure 7 medicina-58-01113-f007:**
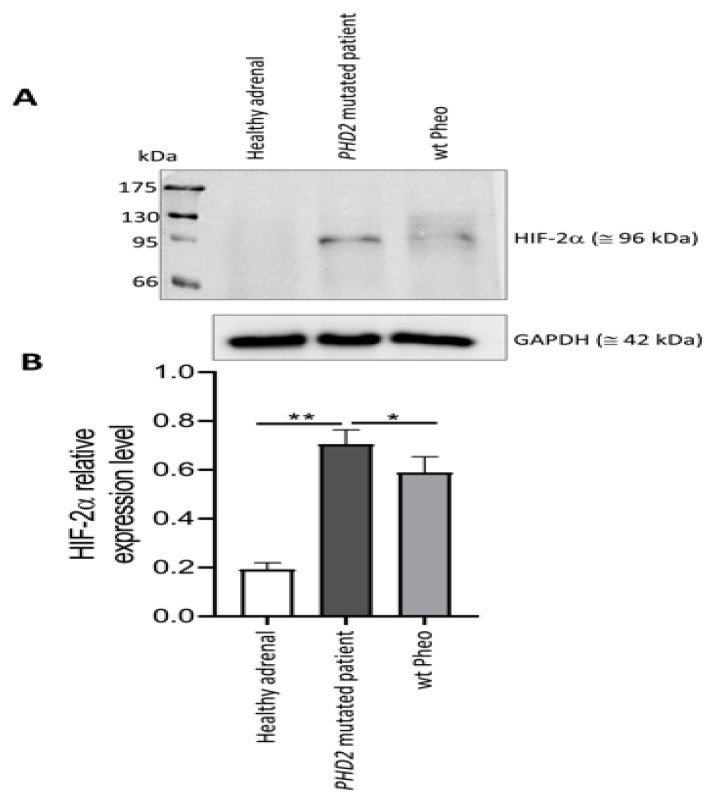
Western blot analysis of the HIFα protein. (**A**): Total protein lysates (40 μg of proteins) of healthy adrenal tissue, tumor specimen from the *PHD2*-mutated patient and a wt Pheo tumor sample were assessed for an HIF2α by Western blot analysis. The blot is representative of three independent preparations; GAPDH immunoblot was used as loading control. (**B**): Densitometric analysis of Western blot immunopositive band intensity: bars represent the mean of intensity ratio between HIF2α and GAPDH in 3 independent blots ± SD. All p values were determined by unpaired one-tailed Student’s *t* test (* *p* < 0.05, ** *p* < 0.01).

## Data Availability

Subject gave her informed consent for inclusion before she participated in the study.

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
