# Peer review of "Novel Germline PHD2 Variant in a Metastatic Pheochromocytoma and Chronic Myeloid Leukemia, but in the Absence of Polycythemia"

_medicina, 2022, doi:10.3390/medicina58081113_

Round 1

Reviewer 1 Report

Provenzano et al submitted an interesting case report of a young female with novel germline PHD2 variant associated with metastatic pheochromocytoma and CML with no polycythemia.  

Major comments:

1- The title should include the presence of CML in the patient. 

2- Data on MCV/MCH/Ferritin were not included in the initial CBC. Co-existing iron deficiency anemia or thalassemia trait may mask the presence of  polycythemia. 

3- Any explanation for the rapid increase in platelets during PRRT and was a work up done to search for myeloproliferative disorder as PRRT may cause thrombocytopenia not thrombocytosis.  

4- There is no discussion in the case description or in the discussion section, whether CML could have been therapy-related post PRRT. 

5- Any explanation for the strong PHD2 positivity in the endothelial cells lining intratumoral capillaires despite the presence of germline mutation?. PHD2 mutation is suggested to be germline, so the absence of PHD2 expression in the tumor might be due to another hit in the other allele. 

6- The discussion part covers important case-related observations and review of literature. However, the paragraphs are fragmented and not easy to follow. For example, the first sentence in the discussion is not clear.  So the discussion needs re-organization. Again there is a need to discuss the possibility of therapy-related CML. 

Minor comments: 

1- The introduction is long with too many references (54 references), it can be summarized to highlight background related to the case instead of the  PPGL in general.

2- The details of methods and materials such as gene panel, SNP-CGH array, WES, analysis, variant prediction, Western blot, RNA, and RT-PCR can be added as supplementary instead of the main manuscript.

 3- Figures 1, 4, and 6 are of low resolution, need to submit high resolution. 

4- It will be helpful to add a schematic of PHD2 gene and site of mutation/domain to Figure 4. 

Author Response

Reviewer 1:

Provenzano et al submitted an interesting case report of a young female with novel germline PHD2 variant associated with metastatic pheochromocytoma and CML with no polycythemia.  

Major comments:

  • The title should include the presence of CML in the patient. 

We added chronic myeloid leukemia according with your suggestion.

  • Data on MCV/MCH/Ferritin were not included in the initial CBC. Co-existing iron deficiency anemia or thalassemia trait may mask the presence of polycythemia. 

We agree with you. We added MCV, MCH and ferritin value in the text (lines 137-139).

  • Any explanation for the rapid increase in platelets during PRRT and was a work up done to search for myeloproliferative disorder as PRRT may cause thrombocytopenia not thrombocytosis.  

Yes, you are right. We checked the blood count for the possible occurrence of thrombocytopenia due to PPRT therapy. At the assay we found thrombcytosis and subsequent investigations led to the diagnosis of CML. In the literature there are no data regarding the increase of platelets during PPRT.

  • There is no discussion in the case description or in the discussion section, whether CML could have been therapy-related post PRRT. 

Due to the presence of chromosome Philadelphia the patient presented the predisposition to develop CML. In our opinion there is no relationship between PPRT therapy and CML.

  • Any explanation for the strong PHD2 positivity in the endothelial cells lining intratumoral capillaires despite the presence of germline mutation? PHD2 mutation is suggested to be germline, so the absence of PHD2 expression in the tumor might be due to another hit in the other allele. 

PHD2 is ubiquitously expressed, and endothelial cells show intense immunohistochemical staining across tissues. In our case, endothelial cells within the patient tumor served as an internal positive control to validate PHD2 antibody specificity.

Thank you very much for this comment, it is true that we cannot rule out the presence of a second shot in the tumor, we have performed the sequencing of the entire exome and the SNP-CCH array in the blood and DNA of the tumor. Whole exome sequencing allows us to study only the coding and flanking regions, but we cannot have any information on the presence of variants in the non-coding regions or in the regulatory regions that could act as a second hit. Furthermore, we have not conducted methylation studies in tumor tissue which could be another important explanation of the clinical picture. We add this comment in the discussion.

  • The discussion part covers important case-related observations and review of literature. However, the paragraphs are fragmented and not easy to follow. For example, the first sentence in the discussion is not clear.  So the discussion needs re-organization. Again there is a need to discuss the possibility of therapy-related CML. 

Thanks for the suggestion. We re-organized the discussion trying to be more clear.

Minor comments: 

  • The introduction is long with too many references (54 references), it can be summarized to highlight background related to the case instead of the PPGL in general.

Thanks for the suggestion. We deleted many references and cited two reviews (Ref 18 and 19).

  • The details of methods and materials such as gene panel, SNP-CGH array, WES, analysis, variant prediction, Western blot, RNA, and RT-PCR can be added as supplementary instead of the main manuscript.

Thanks for the comment. We decide to added supplementary material section.

Figures 1, 4, and 6 are of low resolution, need to submit high resolution. 

We checked the figures. All are in TIF 300 dpi as requested.

  • It will be helpful to add a schematic of PHD2 gene and site of mutation/domain to Figure 4. 

We added a schematic representation of PHD2 gene in which all known domains are reported and the identified variant is also indicated.

Reviewer 2 Report

In this manuscript the group of Dr Canu provides a well written case report of a novel PHD2 mutation in a Phocromocitoma patient. They also illustrate a filtering flowchart to identify harmful genetic variants.

Minor comments:

- The authors should proofread the review, as there are a few minor typographical and grammar errors. Please check that the symbols of genes and proteins are respectively in italics and in capital letters.

Kind Regards

Author Response

Reviewer 2:

In this manuscript the group of Dr Canu provides a well written case report of a novel PHD2 mutation in a Phecromocitoma patient. They also illustrate a filtering flowchart to identify harmful genetic variants.

Minor comments:

- The authors should proofread the review, as there are a few minor typographical and grammar errors. Please check that the symbols of genes and proteins are respectively in italics and in capital letters.

Thanks for the comment. We corrected the typing errors and checked the gene and protein symbols.